# The morphometrical and topographical evaluation of the superior gluteal nerve in the prenatal period

**Alicja Kędzia[1], Krzysztof Dudek[2], Marcin Ziajkiewicz[1], Michal Wolanczyk[1], Anna Seredyn[1], Wojciech Derkowski[3]\*, Zygmunt Antoni Domagala[1]**

1 Wrocław Medical University, Wrocław, Poland, 2 Faculty of Mechanical Engineering, Wrocław University of Science and Technology, Wrocław, Poland, 3 Faculty of Health Sciences, University of Opole, Opole, Poland

\* w.derkowski@hipokrates.org

## Abstract

### Introduction

Advances in medical science are helping to break down the barriers to surgery. In the near future, neonatal or in utero operations will become the standard for the treatment of defects in the human motor system. In order to carry out such procedures properly, detailed knowledge of fetal anatomy is necessary. It must be presented in an attractive way not only for anatomists but also for potential clinicians who will use this knowledge in contact with young patients. This work responds to this demand and presents the anatomy of the superior gluteal nerve in human fetuses in an innovative way. The aim of this work is to determine the topography and morphometry of the superior gluteal nerve in the prenatal period. We chose the superior gluteal nerve as the object of our study because of its clinical significance—for the practice of planning and carrying out hip surgery and when performing intramuscular injections.

### Material and methods

The study was carried out on 40 human fetuses (20 females and 20 males) aged from 15 to 29 weeks (total body length v-pl from 130 to 345 mm). Following methods were used: anthropological, preparatory, image acquisition with a digital camera, computer measurement system Scion for Windows 4.0.3.2 Alpha and Image J (accuracy up to 0.01 mm without damaging the unique fetal material) and statistical methods.

### Results

The superior gluteal nerve innervates three physiologically significant muscles of the lower limb's girdle: gluteus medius muscle, gluteus minimus muscle and tensor fasciae latae muscle. In this study the width of the main trunk of the nerve supplying each of these three muscles was measured and the position of the nerve after leaving the suprapiriform foramen was observed. A unique typology of the distribution of branches of the examined nerve has been created. The bushy and tree forms were distinguished. There was no correlation between the occurrence of tree and bushy forms with the body side (p > 0.05), but it was

**Data Availability Statement:** All relevant data are within the paper.

**Funding:** The study was financed from the statutory funds of the Medical University of

Wrocław. These funds come from a specific grant awarded by the Ministry of Education and Science in Poland to conduct scientific research to maintain the scientific potential of the unit. In the Acknowledgements subsection of the manuscript, the authors included appropriate information about the source of funding required by the grant funder. The funders had no role in study design, data collection and analysis, decision to publish, or preparation of the manuscript. None of the authors of the paper received any salary from the statutory grant (subsidy).

**Competing interests:** The authors have declared that no competing interests exist.

shown that the frequency of the occurrence of the bushy form in male fetuses is significantly higher than in female fetuses ($p < 0.01$). Proportional and symmetrical nerve growth dynamics were confirmed and no statistically significant sexual dimorphism was demonstrated ($p > 0.05$).

## Conclusions

The anatomy of the superior gluteal nerve during prenatal period has been determined. We have identified two morphological forms of it. We have observed no differences between right and left superior gluteal nerve and no sexual dimorphism. The demonstrated high variability of terminal branches of the examined nerve indicates the risk of neurological complications in the case of too deep intramuscular injections and limits the range of potential surgical interventions in the gluteal region. The above research may be of practical importance, for example for hip surgery.

## Introduction

The superior gluteal nerve (*nervus gluteus superior*) is an important element of the sacral plexus. Its fibers come out of branches of the abdominal spinal nerves L4, L5 and S1 [1]. The nerve leaves the pelvis through the suprapiriform foramen, which is the upper part of the greater sciatic opening (*foramen ischiadicum majus*) [2]. It then runs laterally with the superior gluteal artery between the gluteal muscles to innervate the hip joint as well as the muscles: *m. gluteus medius*, *m. gluteus minimus m. tensor fasciae latae* [3, 4] and, as confirmed by recent studies, also *m. piriformis* [5]. The analysis of available literature showed a dynamically growing number of publications concerning the described nerve. The majority of the most recent papers focuses on the relationship between the more common surgical interventions within the hip joint (femoral neck fractures, arthroplastic procedures, hip reconstruction procedures) and damage to the nerves passing through the greater sciatic opening [6, 7]. Due to the nature of the surgical approach to the hip joint, the most frequently damaged structures are the superior gluteal nerve and the superior gluteal artery [1].

Injury of the superior gluteal nerve is a complication of hip surgery that must still be reckoned with. The variability of the course of this nerve, which was studied on dissection material in adults, has a major influence on the risk associated with it [1, 8]. Acute superior gluteal nerve injury following lateral hip arthroplasty, assessed by EMG occurs in 11% to 77% of patients, but is usually temporary and most of these patients' electromyograms recover completely after 3 months. However, in 11% of patients it persists after 9 months, and in one in 12 patients even after a year. In patients who underwent hip arthroplasty with a modified lateral approach, 4 out of 72 had chronic damage of the superior gluteal nerve [8]. Electromyographic examination (EMG) together with electroneurography (ENG) allows for an objective assessment of the degree of nerves and muscles damage in the examined patients [9]. The conclusion of these tests was, inter alia, the suggestion of performing an ultrasound examination before starting the procedure [1]. These studies, apart from their clinical significance, are of course also scientific, providing a more adequate description of the anatomical course of this nerve. No similar research on fetuses to understand the development of this nerve has been performed so far.

The process of limb formation in the embryonic period is relatively well known. Recently, the factors responsible for the location of the limbs, their polarity and identity have been identified [10]. The Pitx1 gene has been found to be the main marker of symmetry and identity of

hind limbs in animals [11]. However, there is no precise data related to the development of innervation of limb muscles. Moreover, there are no studies describing the division of peripheral nerve branches and explaining the cause of the variability of nerve branches observed in further stages of ontogenesis. Some authors point out the high variability of sartorius muscle in the prenatal period, which allows to assume that morphological variants are congenital [12].

It has been demonstrated that the development of the lower limb consists mainly of its growth [13–15].

When analyzing the variability of the distribution of lower limb nerve branches, there are many literature papers on the following nerves: sciatic, common fibular or various cutaneous nerves [13, 16–18]. On the other hand, there is no detailed description of gluteal nerves in the prenatal period in the available literature.

There have been more and more surgical interventions in the perinatal period in recent years, especially in the case of deformities or in the presence of neoplastic lesions involving the lower limb [18].

In the case of newborns or infants, there are practically no scientific papers assessing the morphology of nerves innervating the gluteal region. All clinical recommendations are an extrapolation of data obtained from studies conducted on deceased adults [19]. Precise data on the location and distribution of the branches of these nerves is clinically essential to reduce the number of iatrogenic complications associated with damage to the inferior gluteal, superior gluteal and sciatic nerves. Although the most common injection site in the neonatal period is the lateral surface of the thigh (vastus lateralis muscle), understanding the developmental anatomy of the superior gluteal nerve is of practical importance for planning an injection later in childhood and in adulthood. An injury to the superior gluteal nerve results in the weakening of the limb's abduction movements in the hip joint, the so-called duckling gait or Trendelenburg gait [6, 20]. Their common trauma can cause complete limb paralysis, resulting in massive clinical consequences at childhood [21, 22]. The World Health Organization estimates that around 12 billion injections into the gluteal region are made in connection with the administration of vaccines each year. Nerve damage in this area, including damage to the gluteal nerves, is one of the most common complications of these procedures [19]. The knowledge of the anatomy of this nerve, its development in the prenatal period and anomalies disturbing nerve conduction is therefore not only of cognitive importance but also of very significant clinical importance. Therefore, it is essential to study the details of morphology and topography of superior gluteal in the fetal period.

The aim of this study is to determine the topography and morphometry of the superior gluteal nerve in the prenatal period.

## Material and methods

The study was carried out on human fetuses originating from the collection of the local Department of Anatomy at Wrocław Medical University. The whole material included 40 fetuses (20 females, 20 males) at the age from 15 to 29 weeks of fetal life with a total body length (v-pl, vertex-plantare) ranging from 130 to 345 mm (Table 1).

It was obtained from local obstetric clinics as a result of premature and early childbirths and miscarriages between 1960 and 1996. The fetuses were stored in an appropriate solid preservative solution, containing ethanol, glycerol and formaldehyde in constant proportions, in a room with a permanent temperature.

The manner in which the fetuses were stored has not changed throughout the entire storage period. We have not included in the study fetuses with visible developmental malformations (any malformation in general) and those that did not have complete clinical documentation.

**Table 1. Basic characteristics of the analyzed material.**

| Measurement feature | Female fetuses N = 20 | Male fetuses N = 20 | p-value |
|---|---|---|---|
| Morphological age, t (weeks): | | | |
| Mean ± SD | 21.7 ± 3.4 | 21.8 ± 3.3 | 0.963[a] |
| Median (Q1 –Q3) | 22 (20–24) | 20 (19–24) | |
| Total body length, v-pl (mm) | | | |
| Mean ± SD | 239 ± 57 | 240 ± 51 | 0.981[a] |
| Median (Q1 –Q3) | 250 (213–280) | 219 (205–276) | |
| Crown-rump length, v-tub (mm) | | | |
| Mean ± SD | 169 ± 37 | 169 ± 34 | 0.989[a] |
| Median (Q1 –Q3) | 175 (154–193) | 154 (146–191) | |
| Body mass, m (g) | | | |
| Mean ± SD | 326 ± 195 | 360 ± 239 | 0.628[a] |
| Median (Q1 –Q3) | 315 (182–438) | 241 (182–492) | |

Q1 –lower quartile, Q3 –upper quartile

[a] t–test for independent samples; SD–standard deviation

The value and credibility of the fetal collection has been confirmed in many scientific papers [15, 16, 23–26]. The basic characteristics of the study material are presented in Table 1. There were no statistically significant differences in somatic features of fetuses of both sexes (p > 0.05).

The following methods were used: anthropological method, preparatory method, image acquisition with a digital camera, computerized measurement system Scion for Windows 4.0.3.2 Alpha and Image I and statistical methods. These methods were also used in earlier publications by the authors' team based on research material from the same fetal collection [27, 28].

The anthropological method allowed to determine the biological age of the examined fetuses. The morphological age was assessed using measurable parameters such as total body length (v-pl), crown-rump length (CRL or v-tub) and body weight [29]). The preparation method was based on exposing the superior gluteal nerves on 80 limbs using standard sectional instruments.

Image acquisition was performed using a high-resolution digital camera. The next step was the measurements in Scion Image for Windows 4.0.3.2 Alpha and Image J (National Institute of Mental Health–NIMH; https://imagej.nih.gov). Contrast improvement, sharpening, spatial processing were used to improve image quality. Metrological tests were carried out on the basis of a millimeter scale attached to each photographed specimen.

In the study, the width of the main trunk of the nerve supplying each of these three muscles was measured and the direction of the nerve course was observed after leaving the suprapiri-form foramen.

The results of the measurements were analyzed statistically on the basis of Statistica PL statistical package (Statsoft, Tulsa, USA). The statistical analysis was carried out by an experienced statistician with high mathematical skills documented and confirmed in previous analyses [23, 30].

Variables with a distribution consistent with the normal distribution were analyzed using parametric methods. Student's t-test (t-test) was used to assess the significance of differences between mean values in two independent groups. For quantitative variables, mean values (M), standard deviations (SD), medians (Me) and the lower (Q1) and upper (Q3) quartiles, the lowest (Min) and highest (Max) values were measured. Nominal qualitative variables (sex and

type of nerve branch distribution) in multidirectional tables are presented as numbers (n) and structure indices (%). Relationships between categorized variables were assessed using Fisher's exact test. The relationship between quantitative variables was analyzed by estimating Pearson's r correlation coefficients. The test probability at the level of $p < 0.05$ was assumed as significant, whereas $p < 0.01$ was assumed as highly significant. The measurement results were statistically analyzed with the use of a package of statistical program Statistica v. 13.3 (TIBCO Software Inc.). Continuous quantitative variables were assessed for the consistency of their distribution with the theoretical normal distribution. The test W of Shapiro Wilk was used.

The study was approved by the local bioethics committee (No. 495/2020). The authors declare no conflict of interest.

## Results

The basic characteristics of the material were included in Table 1.

The preparation revealed that the superior gluteal nerve innervates three physiologically relevant muscles of the lower limb's girdle: gluteus medius muscle, gluteus minimus muscle and tensor fasciae latae muscle.

The typology of the distribution of the branches of the examined nerve was outlined. The authors found two types of distribution of the branches of the superior gluteal nerve: a tree form (Figs 1–5) and a bushy form. Within the bushy form, a different arrangement of branches

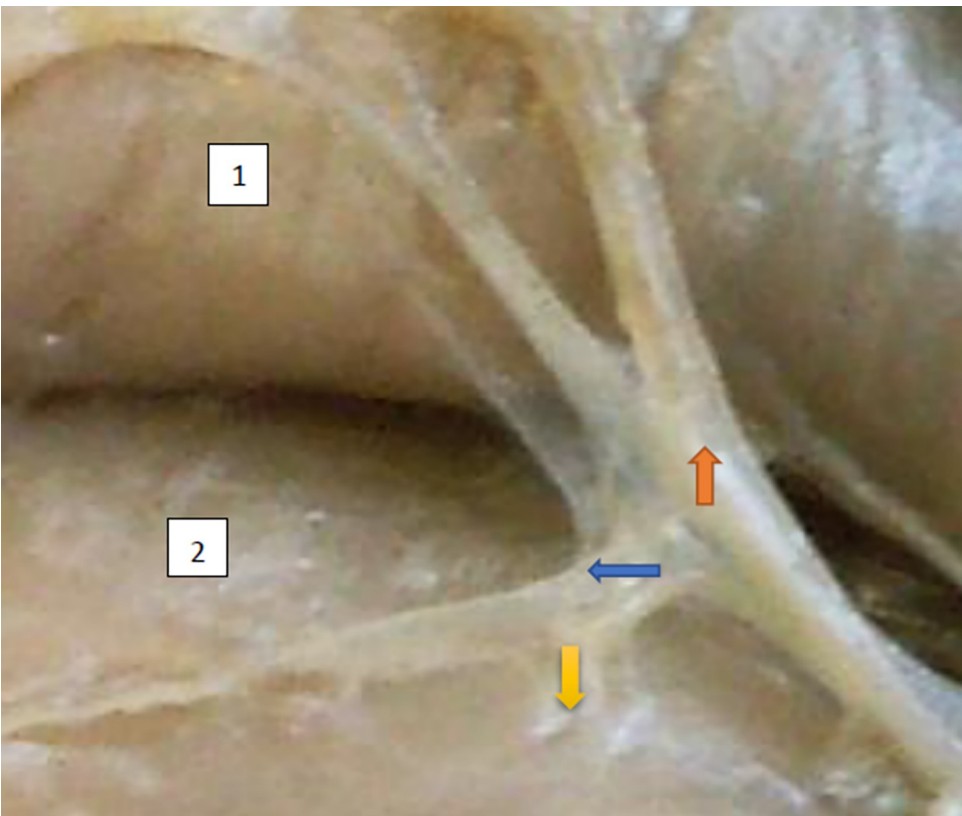

**Fig 1. Superior gluteal nerve, tree type, morphological age 157–23–VI, v–tub 182, v–pl 258, mass 450, female sex, magnification 20x, 1– gluteus medius muscle, 2– gluteus minimus muscle, red arrow–branch gluteus medius muscle, yellow arrow–branch to gluteus minimus muscle, blue arrow–branch to tensor fasiae latae muscle (branch to tensor fasiae latae muscle is better visible in Fig 2).**

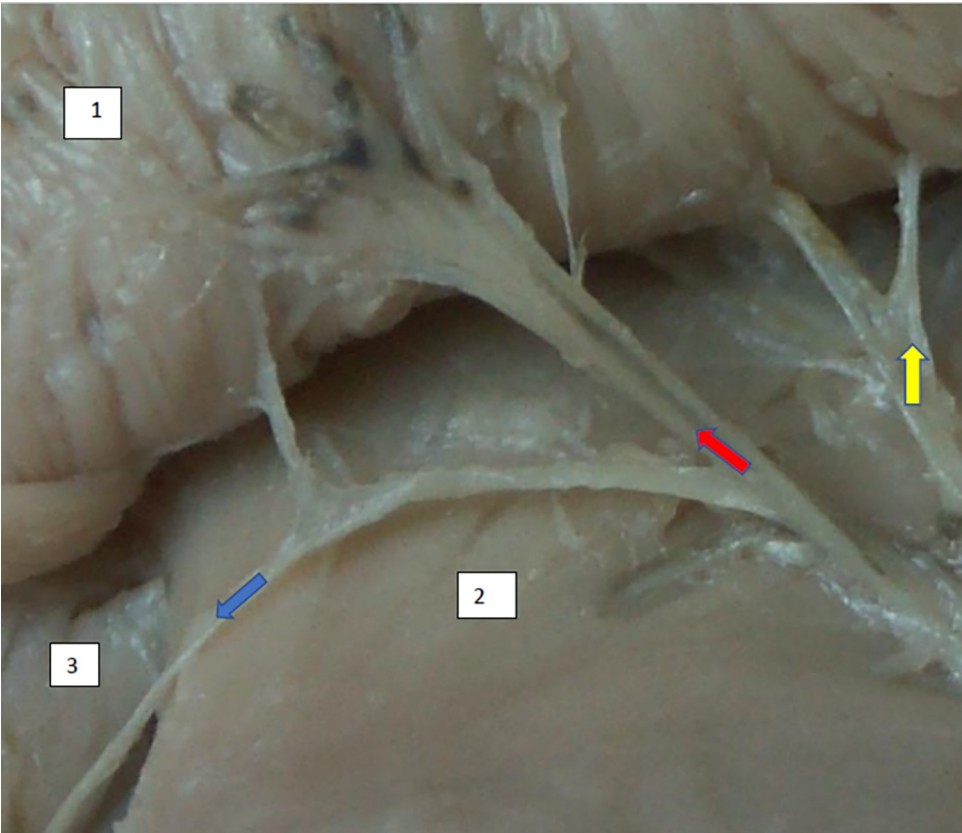

**Fig 2. Superior gluteal nerve, tree type, morphological age 162–24–VI, v–tub 190, v–pl 272, mass 512, male sex, magnification 30x, 1–gluteus medius muscle, 2–gluteus minimus muscle, 3–tensor fasciae latae muscle, red arrow–branch to glueus medius muscle, yellow arrow–another branch to gluteus medius muscle, blue arrow–branch to tensor fasciae latae muscle.**

that reach the muscles was found: vertical (Figs 6–9), horizontal (Figs 10–12). An additional classification is related to the number of secondary branches, which are numerous, with a small diameter of 0.1 mm or sparse in trunks with larger diameters (0.2–0.3 mm). The variety of morphology of the superior gluteal nerve is surprising, from simple, classic forms of a tree, to the connection of two trunks in a bushy form, located vertically or horizontally (within the muscles) forming ladder-shaped formations or nets with various mesh shapes. The preparation difficulties associated with small, easily damaged branches should be emphasized.

Based on these observations a unique typology of the distribution of the branches of the examined nerve was created. The tree (Fig 13) and bushy, predominantly male (Fig 14) forms have been identified.

The frequency of occurrence of different types of nerve on the left and right side and in female and male fetuses was analyzed by comparing them in 2 x 2 contingency tables (Table 2). They show the number (percentage) of fetuses in groups differing in sex and nerve type, the results of the independence tests and the values of the odds ratios and their 95% confidence intervals.

There was no correlation between the occurrence of tree and bushy forms with the body side (p > 0.05), whereas the frequency of occurrence of bushy forms in male fetuses is significantly higher than in female fetuses (p < 0.01). The chance of occurrence of bushy nerve form in male fetuses is six times higher than in female fetuses (OR = 6).

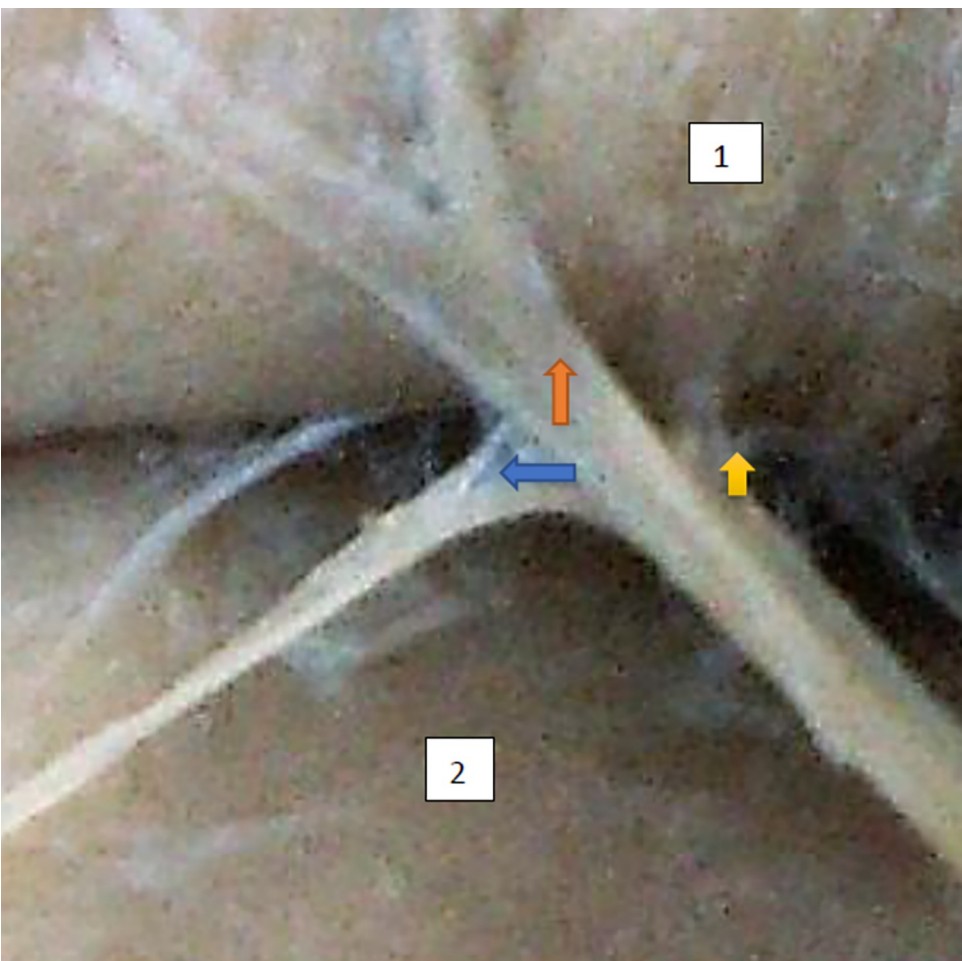

**Fig 3. Superior gluteal nerve, tree type, morphological age 184–27–VII, V–tub 220,V–pl 331.** 124–18–V, V–tub 130,V–pl 190, mass 627, male sex, magnification 40x, –1– gluteus medius muscle, 2– gluteus minimus muscle, red arrow–branch to gluteus medius muscle, yellow arrow–branch to gluteus minimus muscle, blue arrow–branch to tensor fasiae latae muscle.

The dimensions of the examined anatomical structures on both sides are equal (Table 3) and there is no statistically significant sexual dimorphism noted ($p > 0.05$). The results of measurements on the left and right sides and the female and male sexes were combined during the analysis of correlation and regression of the dimensions of the examined structures with the age of the fetus. It was due to demonstrated lack of sexual dimorphism and asymmetry.

The values of linear correlation coefficients are presented in Table 4. In the case of statistically significant correlation ($p < 0.05$), they are also presented in the correlation diagrams, which include regression models (Figs 15–18).

The width of the piriformis muscle in the analysed period increases progressively at a linear rate of 0.22 mm per week and the width of the main branch to the gluteus medius muscle increases at a rate of 0.01 mm per week.

Due to the high variability of the dimensions of the nerve branches running to tensor fasciae latae muscle (TFL), the correlation between the dimensions of the main nerve branch running to gluteus minimus muscle (MINI) and the average width of the branches running away from the trunk of the main nerve (MINI bis), resulting from a high relative measurement error, is not statistically significant.

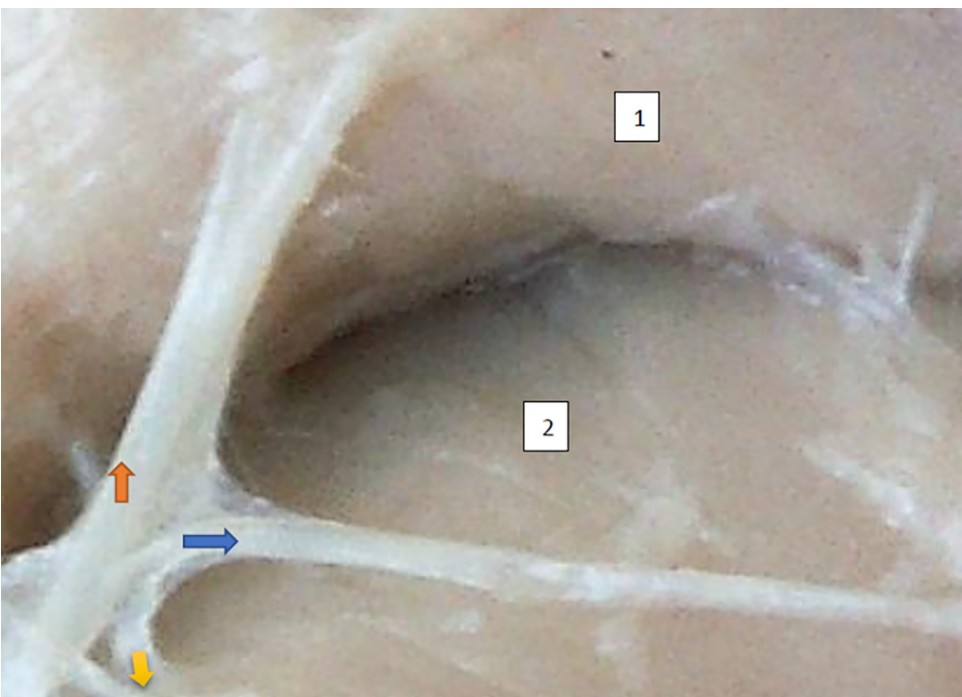

**Fig 4. Superior gluteal nerve, tree type, morphological age 124–18–V, V–tub 130,V–pl 190, mass 140, male sex, magnification 30x.** 1– gluteus medius muscle, 2– gluteus minimus muscle, red arrow–branch to gluteus medius muscle, yellow arrow–branch to gluteus minimus muscle, blue arrow–branch to tensor fasiae latae muscle.

The growth of the nerve is proportional as evidenced by a statistically significant positive correlation of TFL, MP, MED, MINI and MINI bis (Figs 17 and 18).

## Discussion

The course of the superior gluteal nerve and its importance for the safety of surgical access to the hip joint was studied on dissection material in adults [8]. The authors distinguished two types of the course of this nerve: the spray pattern course and the transverse neural trunk pattern course. The course of the superior gluteal nerve was also studied by Jacobs and Buxton, who described two types of the course of this nerve. According to them, the ends of all branches of the nerve assumed an arcuate shape [31]. Perez et al. (2004) examined the dissection material of 19 adults [32]. They found that the superior gluteal nerve was in 89.48% cases divided into two branches and in 10.52% cases into three. The course of the superior gluteal nerve in adults was also studied by Ray et al. (2013). Among other things, they found that the most frequent number of branches to the individual muscles innervated by the superior gluteal nerve ranged from 2 to 3 [1].

Most of the work analyzing the development of the fetus was conducted using in vivo techniques. Various anatomical structures were studied: biparietal diameter, head and body circumferences, transverse brain dimensions, abdomen circumferences, and limb lengths [33–37].

The majority of the literature surrounding the development of the fetus has focused on fetal growth of the vertebral column [38–41]. In one study, authors evaluated the development of the sacral bone using ultrasonography [40], confirming the importance of biparietal diameter measurement in the assessment of physiological pregnancy. Similarly, in this paper, biparietal diameter was also used in preparation analysis to confirm the age of examined fetuses.

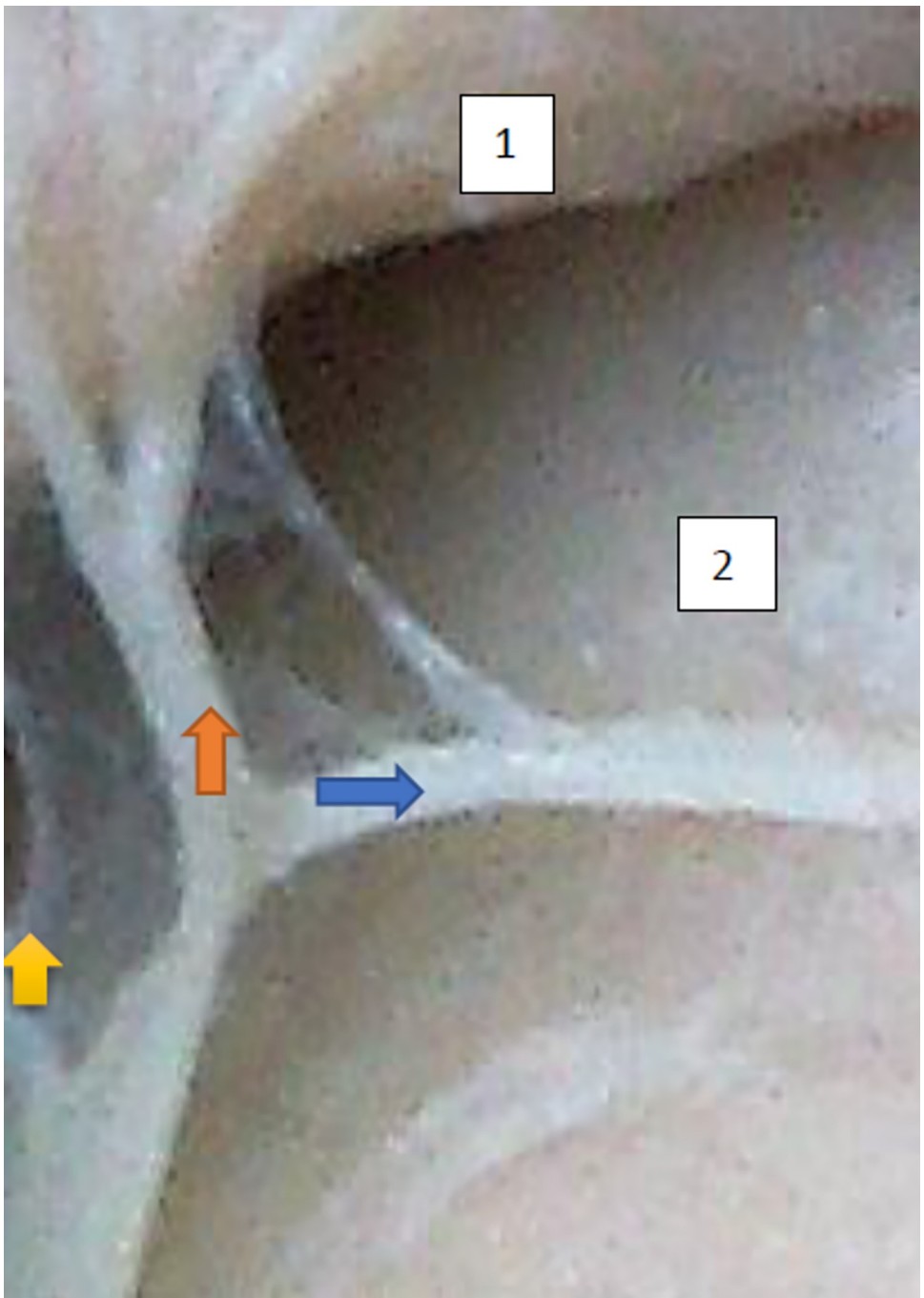

**Fig 5. Superior gluteal nerve, tree type, morphological age150–22–VI, v–tub 182,v–pl 258, mass 140, female sex, magnification x30, 1– gluteus medius muscle, 2– gluteus minimus muscle, red arrow–branch to gluteus medius muscle, yellow arrow–branch to gluteus minimus muscle, blue arrow–branch to tensor fasiae latae muscle.**

Throughout the study period, the scope of information derived from the available literature is mainly related to the analysis of topography and metrology of the superior gluteal nerve in adults [1, 3, 6].

The course and topography of the superior gluteal nerve is of significant importance in developing safe surgical access procedures to the hip joint [42]. Many authors also emphasize

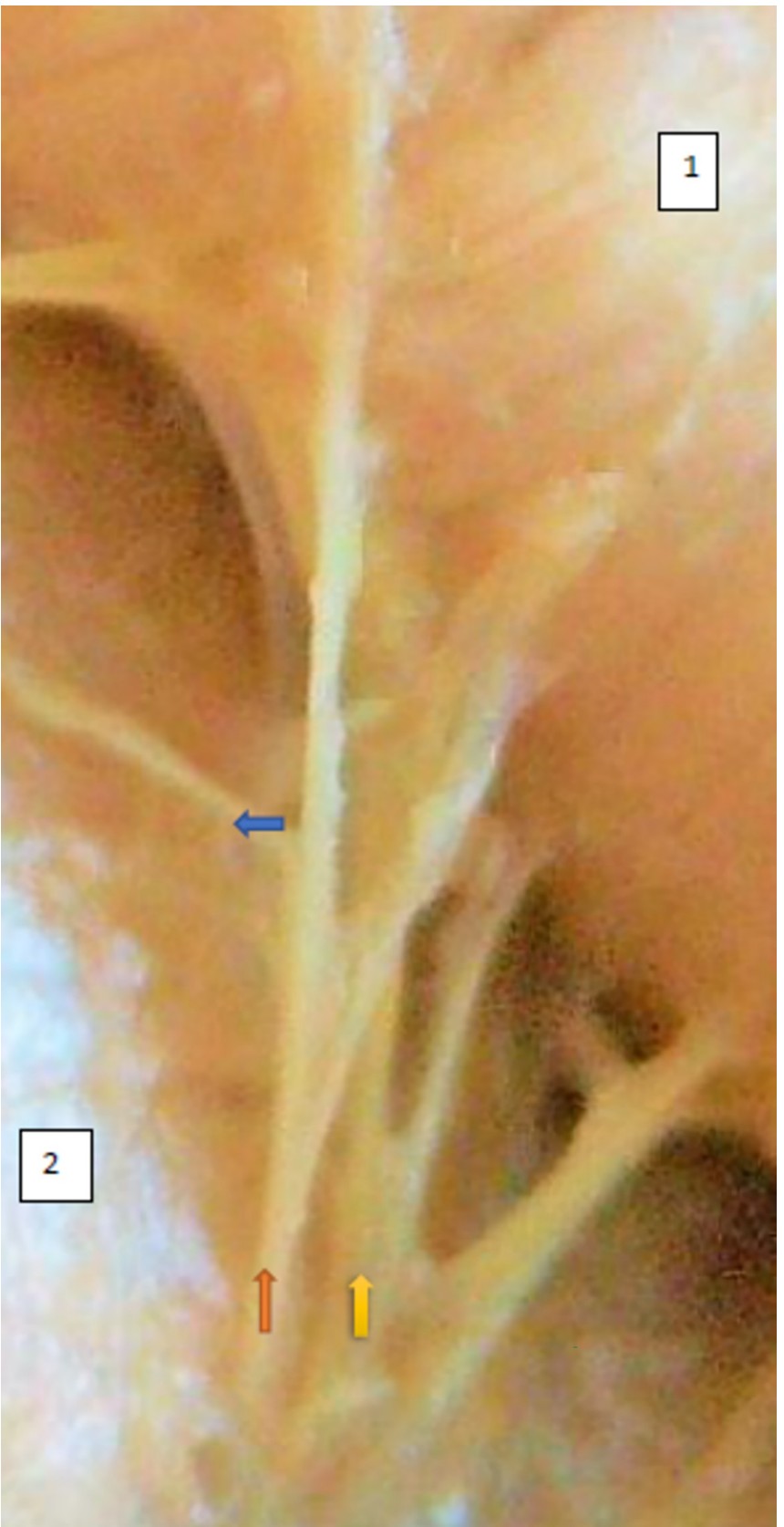

**Fig 6. Superior gluteal nerve bushy type, vertical course morphological age 162–24–VI, v–tub183, v–pl 270,mass 544, male sex, magnification 45x.** 1– gluteus medius muscle, 2– gluteus minimus muscle, red arrow–branch to gluteus medius muscle, yellow arrow–branch to gluteus minimus muscle, blue arrow–branch to tensor fasiae latae muscle.

the high frequency of nerve injuries during pelvic bone fractures and hip dislocations. The nerve may also be the object of stab wound injuries or even the point of iatrogenic injuries resulting from intramuscular injections [43, 44].

With regard to prenatal studies, the available literature is extremely scarce. The only available studies related to the evaluation of the gluteal region concern, in most cases, the assessment of the superior gluteal muscle and analysis of sciatic nerve variability [27, 45].

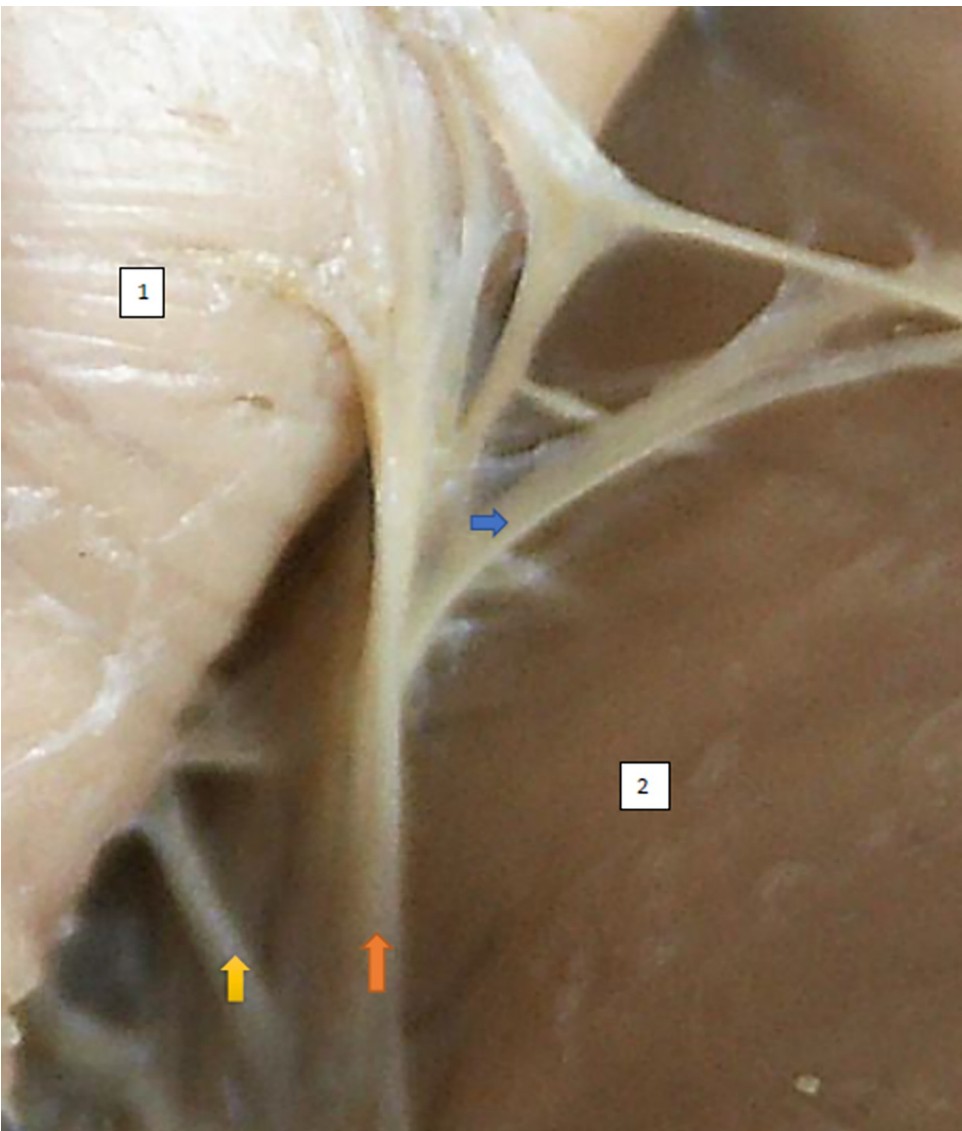

**Fig 7. Superior gluteal nerve bushy type, vertical course, variform meshes of the net on the branch extending to gluteus medius morphological age 195–28–VII, v–tub 235,v–pl 337, mass 842, male sex, magnification 50 x,** 1– gluteus medius muscle, 2– gluteus minimus muscle, red arrow–branch to gluteus medius muscle, yellow arrow–branch to gluteus minimus muscle, blue arrow–branch to tensor fasiae latae muscle.

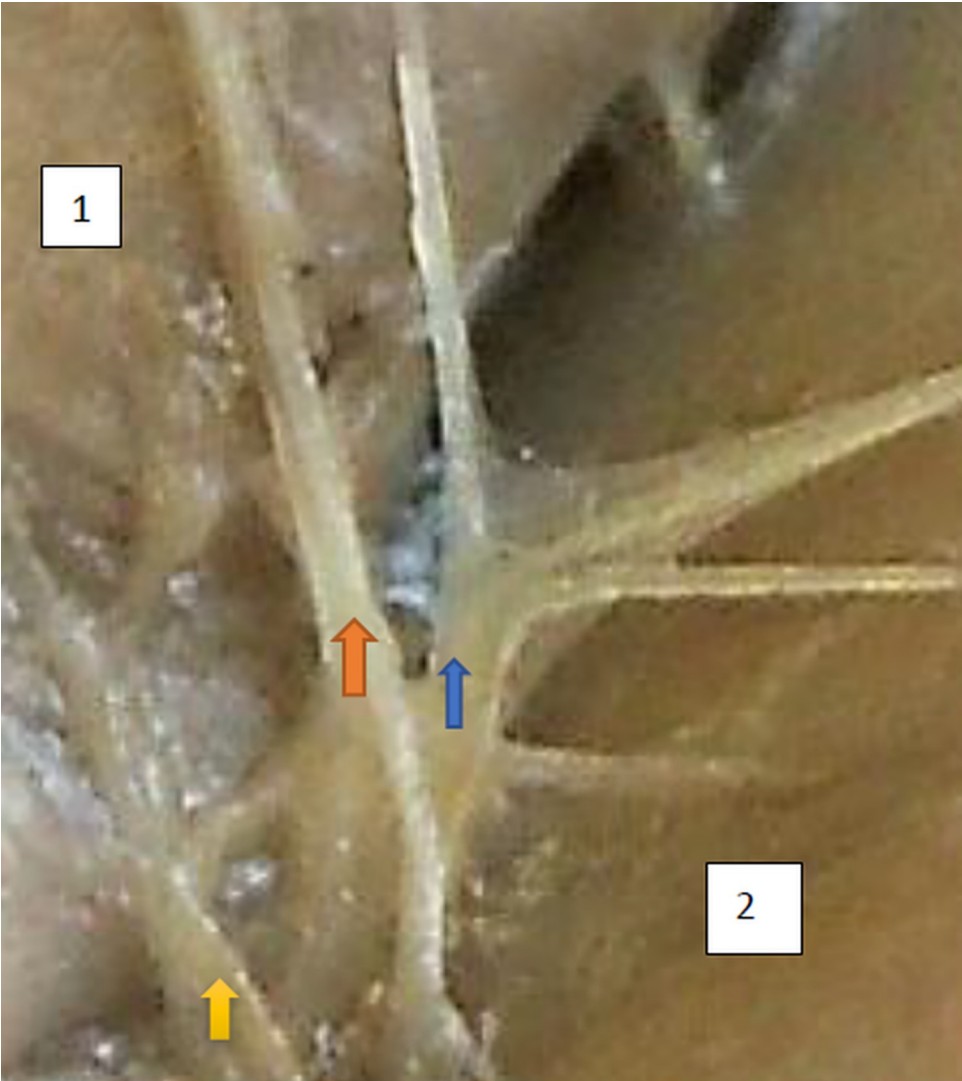

**Fig 8. Superior gluteal nerve bushy type, vertical course, morphological age 157–23–VI, v–tub182, v–pl 258, mass 450, female sex, magnification 20x, 1– gluteus medius muscle, 2– gluteus minimus muscle, red arrow–branch to gluteus medius muscle, yellow arrow–branch to gluteus minimus muscle, blue arrow–branch to tensor fasiae latae muscle.**

Neuromuscular platelets of the gluteal region were analyzed through microscopic examination, which were based on fetal preparations as well as neurovascular pedicles. Microscopic analyses confirmed the high variability of the terminal branches of the superior gluteal nerve in the current study [46, 47].

The dynamics of growth of the examined nerve is proportional and comparable to the dynamics of other nerves analyzed in separate studies [48, 49]. This research results revealed that the width of the piriformis muscle in the analyzed period increases linearly at a rate of 0.22 mm per week, and the width of the main branch to the gluteus medius muscle average at a rate of 0.01 mm per week.

Similar results in terms of developmental dynamics were obtained when comparing piriformis muscle with other analyzed muscles in the fetal period [50]. Attention was drawn to the symmetrical development of the width of the piriformis muscle and the superior gluteal nerve

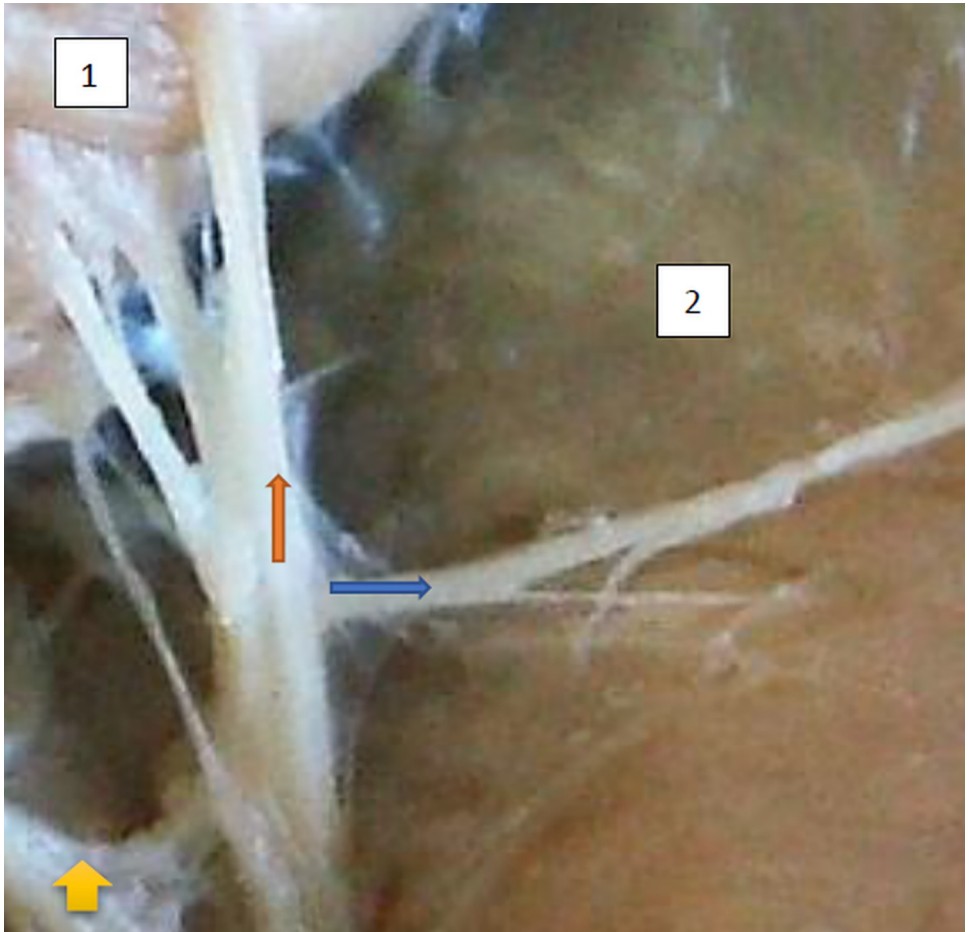

**Fig 9. Superior gluteal nerve bushy type, vertical coursemorphological age 164–24–VI,v–tub191, v–pl 269. mass 489, male sex, magnification 20x, 1– gluteus medius muscle, 2– gluteus minimus muscle, red arrow–branch to gluteus medius muscle, yellow arrow–branch to gluteus minimus muscle, blue arrow–branch to tensor fasiae latae muscle.**

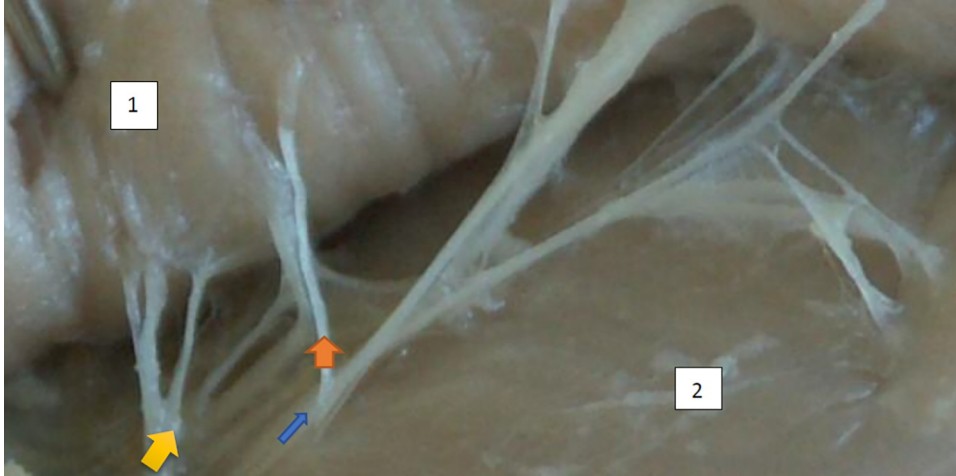

**Fig 10. Superior gluteal nerve, bushy type, horizontal–oblique course, the net with variform meshes morphological age 169–25–VII,V–tub 200, V–pl 294,mass 690, male sex, magnification 20x.** 1– gluteus medius muscle, 2– gluteus minimus muscle, red arrow–branch to gluteus medius muscle, yellow arrow–branch to gluteus minimus muscle, blue arrow–branch to tensor fasiae latae muscle.

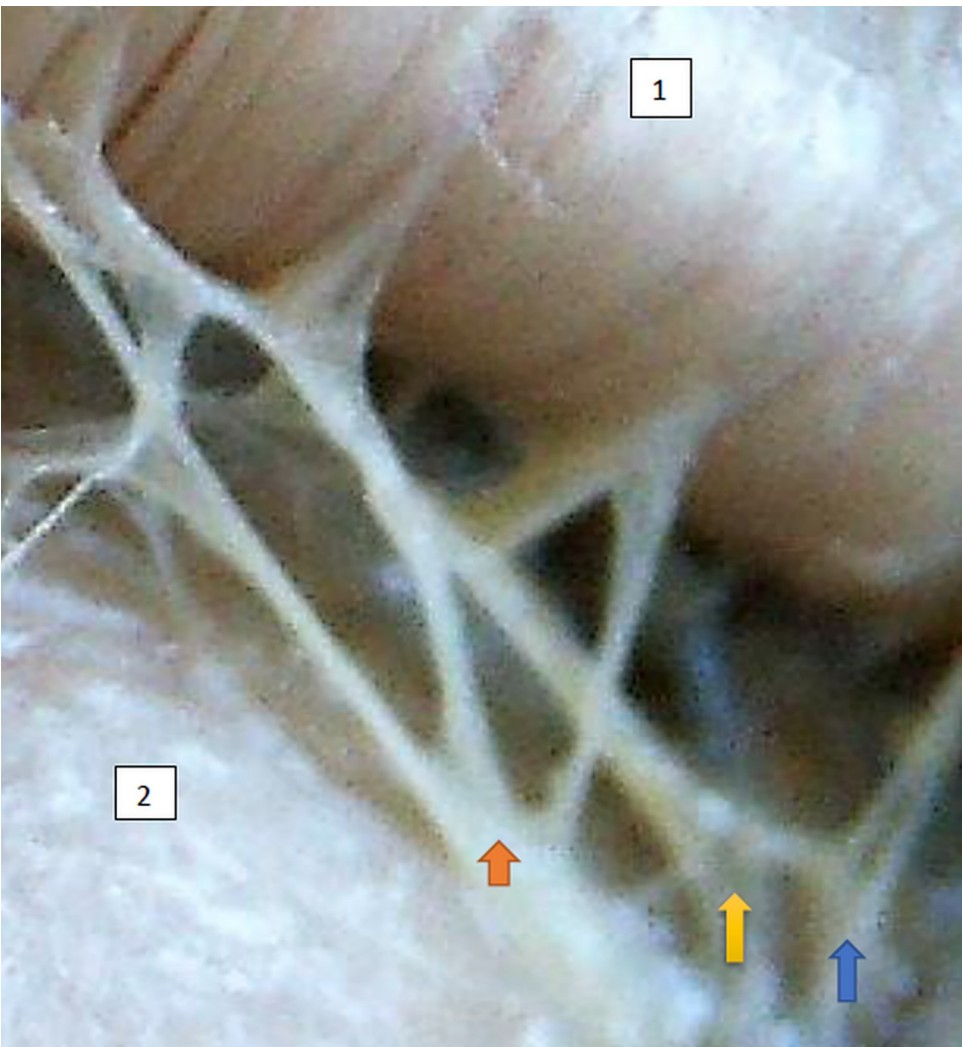

**Fig 11. Superior gluteal nerve, bushy type, horizontal–oblique course, the net with variform meshes, morphological age 169–25–VII,V–tub 200, V–pl 294,mass 690, male sex, magnification 20x.** Figs 10 and 11 are taken from both sides of the same specimen and are evidence of symmetry in the type of innervation., 1– gluteus medius muscle, 2– gluteus minimus muscle, red arrow–branch to gluteus medius muscle, yellow arrow–branch to gluteus minimus muscle, blue arrow–branch to tensor fasiae latae muscle.

as well as the branches diverging from the examined nerve. The aforementioned structures were responsible for the innervation of selected external pelvic muscles. In addition, the lack of sex differences in the analyzed material was also observed. The same characteristics were found in the analysis of development of gluteus maximus muscle [27, 45].

The present study introduces an innovative typology of branch distribution of the superior gluteal nerve, implementing a comprehensive mathematical analysis of the obtained results. These results are of great cognitive importance and can also be used in future research in connection with the dynamically growing specialities of neonatal and intrauterine surgery [51, 52].

Furthermore, the clinical significance of the study of the superior gluteal nerve in the prenatal period is important for the surgical treatment of defects in the lower limbs in infants and possibly also in the fetal period in the future. For example, in the developmental dysplasia of the hip (DDH), which affects approximately one percent of newborns. If it is diagnosed after

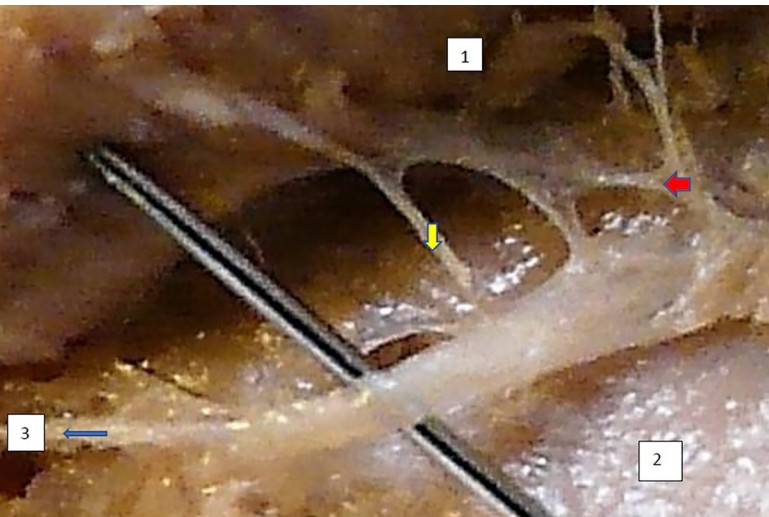

**Fig 12. Superior gluteal nerve, bushy type, horizontal course, morphological age 113–17–V, v–tub 110,V–pl 140, mass 120, male sex, magnification 50x** 1–gluteus medius muscle, 2–gluteus minimus muscle, 3–tensor fasciae latae muscle, red arrow–branch to gluteus medius muscle, yellow arrow–branch to gluteus minimus muscle, blue arrow–branch to tensor fasiae latae muscle.

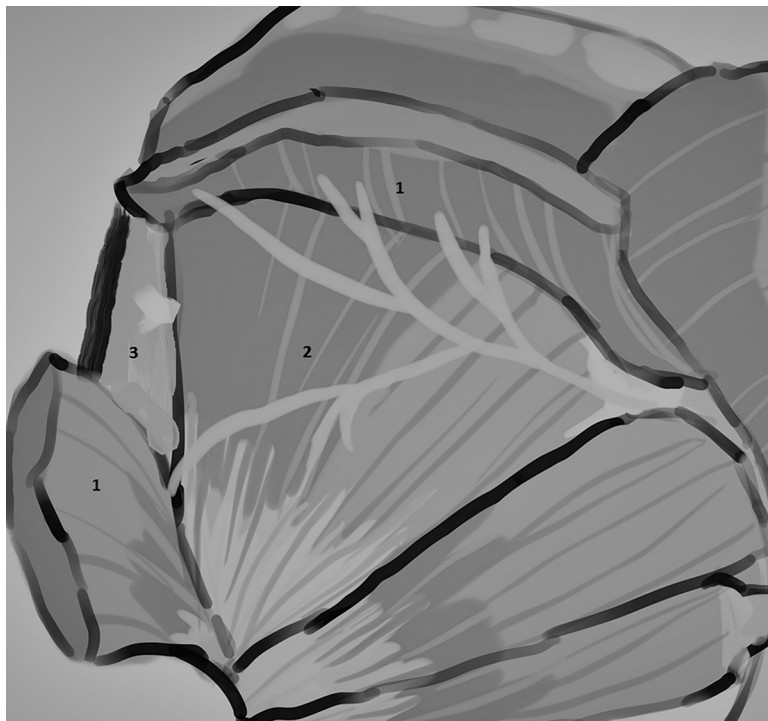

**Fig 13. Superior gluteal nerve, tree type–anatomical diagram of the nerve path (thanks to Julia Derkowska),** 1–gluteus medius muscle, 2–gluteus minimus muscle, 3–tensor fasciae latae muscle.

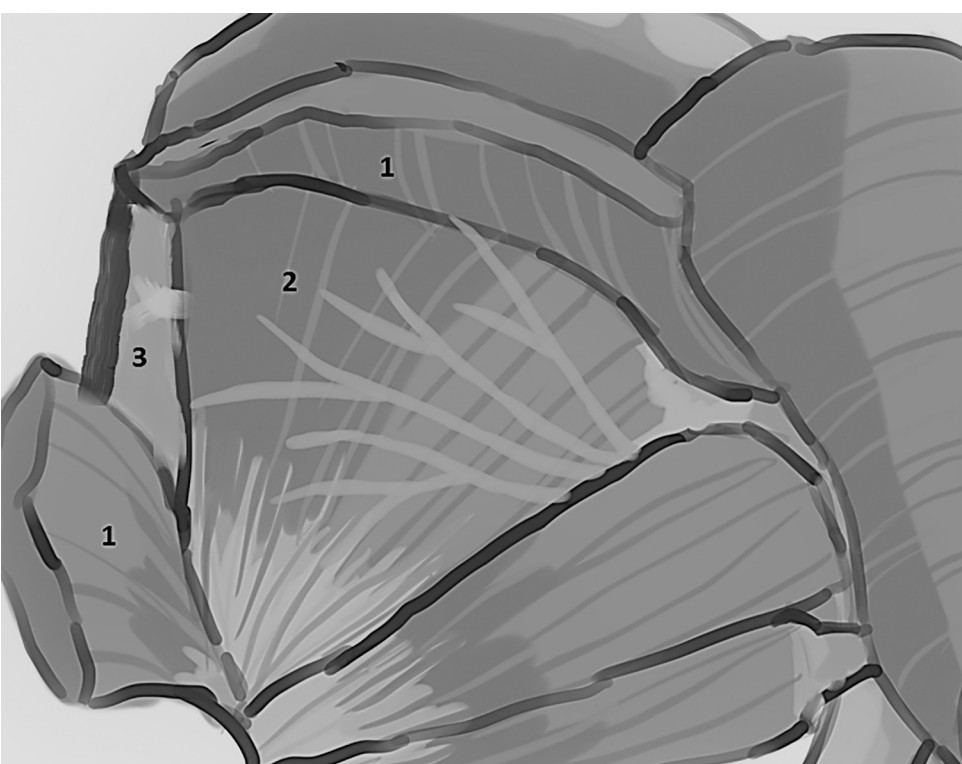

**Fig 14. Superior gluteal nerve, bushy type–anatomical diagram of the nerve path (thanks to Julia Derkowska), 1–gluteus medius muscle, 2–gluteus minimus muscle, 3–tensor fasciae latae muscle.**

the child is six months old, permanent deformation of the tissues of the hip joint may develop. The older the child, the more likely it will be to openly reduce it, with osteotomy of the femur or acetabulum to stabilize reduction. Late open reduction score is significantly inferior to long-term hip function and increases the risk of developing coxarthrosis [53].

The management of developmental dysplasia of the hip depends on age. The sooner neonatal hip instability is diagnosed and treated, the better. Initially, the procedure is based on the

**Table 2. Prevalence of superior gluteal nerve patterns in respect to sex and laterality.** The bold and italic text shows a statistically significant difference.

| Type | Gender | | F vs. M p-value* | OR (95% CI) |
|---|---|---|---|---|
| | Female (F) *N* = 20 | Male (M) *N* = 20 | | |
| Nerve pattern on the left: | | | | |
| Tree | 8 (40%) | 2 (10%) | 0.068 | ***6.00 (1.08–33.3)*** |
| Bushy | 12 (60%) | 18 (90%) | | 1.00 (ref.) |
| Nerve pattern on the right: | | | | |
| Tree | 8 (40%) | 2 (10%) | 0.068 | ***6.00 (1.08–33.3)*** |
| Bushy | 12 (60%) | 18 (90%) | | 1.00 (ref.) |
| Nerve pattern on both sides: | | | | |
| Tree | 16 (40%) | 4 (10%) | ***0.005*** | ***6.00 (1.79–20.1)*** |
| Bushy | 24 (60%) | 36 (90%) | | 1.00 (ref.) |

* Pearson's Chi–squared test with Yates' continuity correction

**Table 3. Width of the superior gluteal nerve branches to the particular muscles in respect to sex and laterality.**

| Measurement feature | Gender | | F vs. M p-value |
|---|---|---|---|
| | Female (F) $N$ = 20 | Male (M) $N$ = 20 | |
| *MP L* [mm]–MP width, left side | | | 0.995a |
| Mean ± SD | 4.2 ± 1.3 | 4.2 ± 1.2 | |
| Median (Q1 –Q3) | 4.1 (3.1–5.3) | 4.0 (3.2–4.6) | |
| *MP R* [mm]–MP width, right side | | | 0.848a |
| Mean ± SD | 4.2 ± 1.3 | 4.2 ± 1.2 | |
| Median (Q1—Q3) | 4.1 (1.9–6.5) | 4.0 (2.8–6.8) | |
| *MP L* vs. *MP R* | $p$ = 0.311b | $p$ = 0.558b | |
| *TFL L* [mm]–the width of the nerve branch to TFL, left side | | | 0.428a |
| Mean ± SD | 0.26 ± 0.03 | 0.27 ± 0.05 | |
| Median (Q1 –Q3) | 4.1 (3.1–5.3) | 4.0 (3.4–4.6) | |
| *TFL R* [mm]–the width of the nerve branch to TFL, right side | | | 0.998a |
| Mean ± SD | 0.26 ± 0.05 | 0.26 ± 0.04 | |
| Median (Q1 –Q3) | 0.25 (0.24–0.28) | 0.26 (0.24–0.30) | |
| *TFL L* vs. *TFL R* | $p$ = 0.388b | $p$ = 0.180b | |
| *MED L* [mm]–width of main nerve branch to MED, left side | | | 0.758a |
| Mean ± SD | 0.25 ± 0.05 | 0.25 ± 0.06 | |
| Median (Q1 –Q3) | 0.25 (0.20–0.29) | 0.25 (0.20–0.30) | |
| *MED R* [mm]–width of main nerve branch to MED, right side | | | 0.691a |
| Mean ± SD | 0.24 ± 0.05 | 0.25 ± 0.06 | |
| Median (Q1 –Q3) | 0.24 (0.21–0.27) | 0.25 (0.20–0.29) | |
| *MED L* vs. *MED R* | $p$ = 0.091b | $p$ = 0.126b | |
| *MINI L* [mm]–width of main nerve branch to MINI, left side | | | 0.139a |
| Mean ± SD | 0.22 ± 0.04 | 0.24 ± 0.05 | |
| Median (Q1 –Q3) | 0.21 (0.19–0.25) | 0.26 (0.19–0.29) | |
| *MINI R* [mm]–width of main nerve branch to MINI, right side | | | 0.193a |
| Mean ± SD | 0.22 ± 0.05 | 0.24 ± 0.05 | |
| Median (Q1 –Q3) | 0.22 (0.19–0.25) | 0.24 (0.20–0.29) | |
| *MINI L vs. MINI R* | $p$ = 0.762b | $p$ = 0.322b | |
| *MINI bis L* [mm]–the average width of nerve branches exiting the nerve trunk on the left side | | | 0.606a |
| Mean ± SD | 0.10 ± 0.02 | 0.11 ± 0.02 | |
| Median (Q1 –Q3) | 0.10 (0.09–0.12) | 0.10 (0.10–0.12) | |
| *MINI bis R* [mm]–the average width of nerve branches exiting the nerve trunk on the right side | | | 0.958a |
| Mean ± SD | 0.11 ± 0.02 | 0.11 ± 0.03 | |
| Median (Q1 –Q3) | 0.10 (0.10–0.13) | 0.12 (0.10–0.13) | |
| *MINI bis L* vs. *MINI bis R* | $p$ = 0.149b | $p$ = 0.937b | |

Q1 –lower quartile, Q3 –upper quartile, a t–test for independent samples; b t–test for dependent samples.

a t–test for independent samples

b t–test for dependent samples. MP–piriformis muscle; TFL–tensor fasciae latae muscle; MED–gluteus medius muscle; MINI–gluteus minimus muscle.

use of a device that holds the hip in the abduction. In the event of ineffectiveness of such a procedure, a surgical reduction is indicated [54].

Limitations of the study were two: the quantitative limitation and the limitation resulting from the nature of the fetal material. The quantitative limitation of the study was due to the fact that fetal material is currently particularly difficult to obtain, and obtaining fetuses for anatomical research is a big challenge. The limitation resulting from the nature of the material was due to specific features of fetal tissues distinguishing them from adult cadavers' tissues. This

**Table 4. The correlation coefficients of the analyzed parameters with the age of the fetus.**

| Measuring feature | Correlation coefficient | Level of significance |
|---|---|---|
| MP (mm) | $r = 0.608$ | ***p < 0.001*** |
| TFL (mm) | $r = 0.177$ | $p = 0.117$ |
| MED (mm) | $r = 0.330$ | ***p = 0.003*** |
| MINI (mm) | $r = 0.108$ | $p = 0.117$ |
| MINI bis (mm) | $r = 0.077$ | $p = 0.520$ |

MP–width of piriformis muscle, TFL–width of the main branch to tensor fasciae latae muscle, MED–width of main branch to gluteus medius muscle, MINI–width of main nerve branch to gluteus minimus muscle, MINIbis–the average width of nerve branches exiting the nerve trunk of the nervus gluteus superior

requires from the anatomist much more precision, and often special techniques for the preservation, storage, and examination of the fetuses.

## Conclusions

The morphology of the superior gluteal nerve has been determined. The bushy and tree forms have been identified. No lateralisation of individual types was observed. No sexual dimorphism and asymmetry were found.

This study has a significant clinical importance for planning and performing hip surgery and intramuscular injections in the gluteal area. It could help avoid the damages made to the superior gluteal nerve and mitigate the risks related to such surgeries. They are now among the most frequently made damages because of the nature of surgical access to the hip joint.

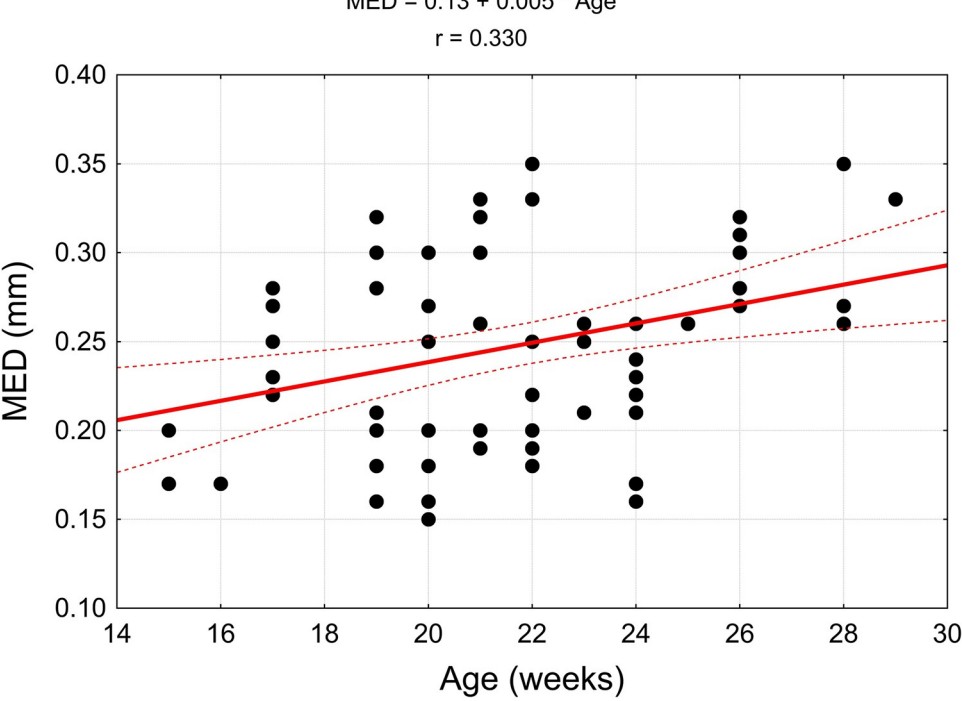

**Fig 15. Correlation diagrams for the width of the piriformis muscle (MP) and width of the main branch to the gluteus medius muscle (MED) with the age of the fetus.**

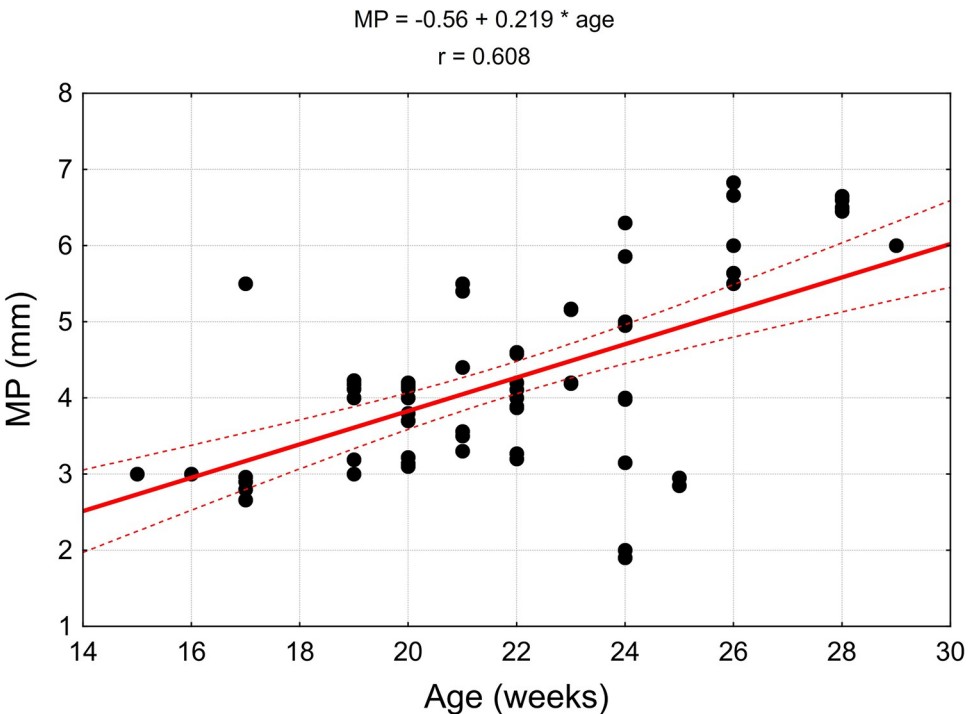

**Fig 16. Correlation diagrams for the width of the piriformis muscle (MP) and width of the main branch to the gluteus medius muscle (MED) with the age of the fetus.**

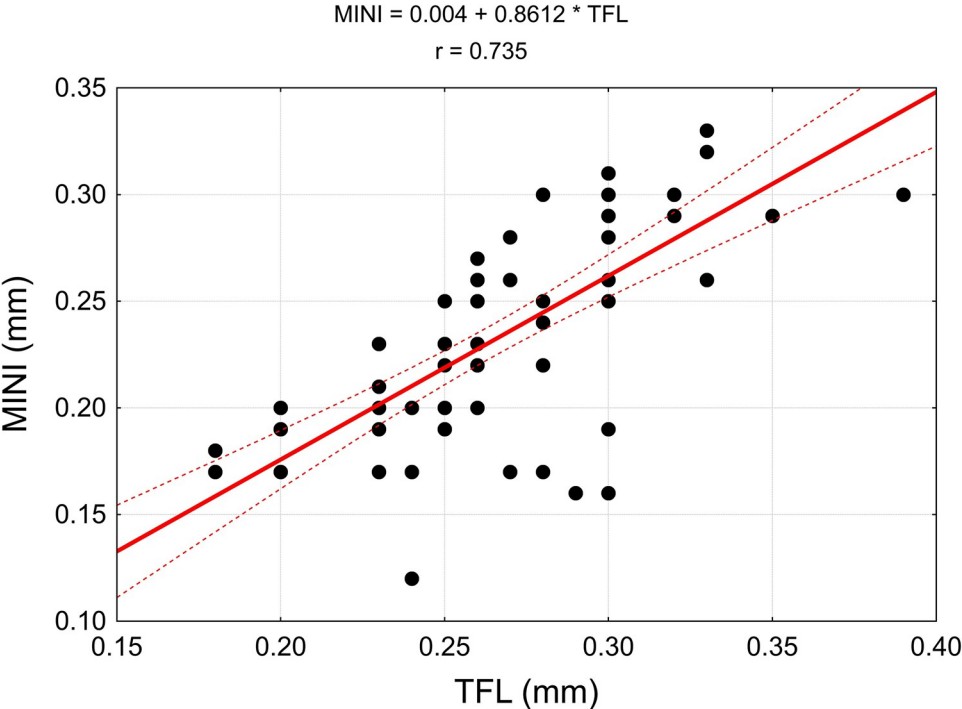

**Fig 17. Correlation diagrams for the width of the nerve branch to the tensor fasciae latae muscle (TFL) with the width of the piriformis muscle (MP), the width of the main branch to the gluteus medius muscle (MED), the average width of the branches emerging from the main trunk of the nerve to the gluteus minimus muscle (MINI) and the average total width of the branches departing from the main trunk of the examined nerve (MINI bis).**

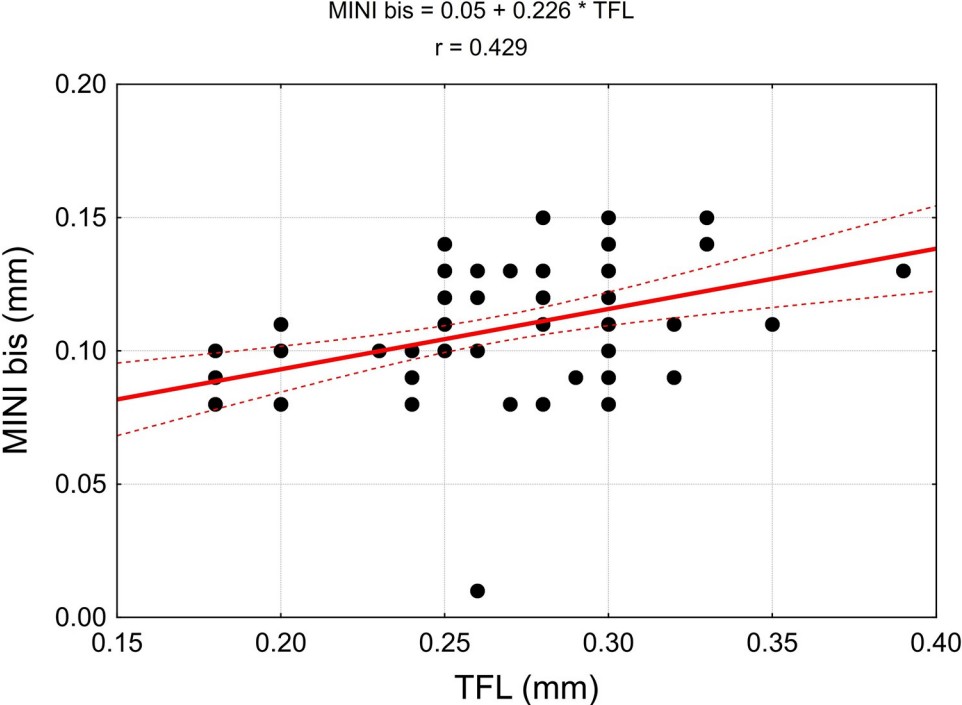

**Fig 18. Correlation diagrams for the width of the nerve branch to the tensor fasciae latae muscle (TFL) with the width of the piriformis muscle (MP), the width of the main branch to the gluteus medius muscle (MED), the average width of the branches emerging from the main trunk of the nerve to the gluteus minimus muscle (MINI) and the average total width of the branches departing from the main trunk of the examined nerve (MINI bis).**

What is more, the clinical importance of our study of the superior gluteal nerve in the prenatal period is of great importance for the surgical treatment of lower limb defects in infants, and possibly also in the fetal period in the future (for example for developmental dysplasia of the hip—DDH).

## Acknowledgments

The authors sincerely thank those who donated their bodies to science so that anatomical research could be performed. Results from such research can potentially increase mankind's overall knowledge that can then improve patient care. Therefore, these donors and their families deserve our highest gratitude.

The authors sincerely thank the statistician—Krzysztof Dudek, B.Eng., M.Sc., Ph.D. for his wonderful work of mastering the intricacies of the statistical and mathematical analysis of our research.

The presented research results, carried out within the framework of the topic according to the register in the S system with the number SUBZ.A351.22.038, were financed from the subsidy granted by the Minister of Science and Higher Education.

Many thanks to Mr. Mirosław Łukaszun for help in choosing the appropriate fetal material. Many thanks also to Victoria Tarkowski, BA from Toronto, Canada and Barbara Podhajska (nee Derkowska) from Wrocław, London for linguistic correction.

Authors declare no conflict of interest.

## Author Contributions

**Conceptualization:** Alicja Kędzia, Krzysztof Dudek.

**Data curation:** Alicja Kędzia, Zygmunt Antoni Domagala.

**Formal analysis:** Alicja Kędzia, Krzysztof Dudek, Zygmunt Antoni Domagala.

**Funding acquisition:** Alicja Kędzia, Zygmunt Antoni Domagala.

**Investigation:** Alicja Kędzia, Marcin Ziajkiewicz, Michal Wolanczyk, Anna Seredyn, Zygmunt Antoni Domagala.

**Methodology:** Alicja Kędzia, Krzysztof Dudek.

**Project administration:** Alicja Kędzia, Zygmunt Antoni Domagala.

**Resources:** Alicja Kędzia, Zygmunt Antoni Domagala.

**Software:** Alicja Kędzia, Krzysztof Dudek.

**Supervision:** Alicja Kędzia.

**Validation:** Alicja Kędzia, Krzysztof Dudek.

**Visualization:** Alicja Kędzia.

**Writing – original draft:** Alicja Kędzia, Krzysztof Dudek, Zygmunt Antoni Domagala.

**Writing – review & editing:** Alicja Kędzia, Wojciech Derkowski.

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
