## [Decision Letter · Decision Letter 0]

15 Mar 2022

PONE-D-21-33849The morphometrical and topographical evaluation of the superior gluteal nerve in the prenatal period.PLOS ONE

Dear Dr. Derkowski,

Thank you for submitting your manuscript to PLOS ONE. After careful consideration, we feel that it has merit but does not fully meet PLOS ONE’s publication criteria as it currently stands. Therefore, we invite you to submit a revised version of the manuscript that addresses the points raised during the review process. In my opinion this manuscript has great scientific potencial, however need to be thoroughly edited according to journal style. Please follow strictly Reviewer 1.

We look forward to receiving your revised manuscript.

Kind regards,

Mateusz Koziej, MD, PhD, DSc

Academic Editor

PLOS ONE

Journal Requirements:

NO

NO

Reviewers' comments:

Reviewer's Responses to Questions

**Comments to the Author**

1. Is the manuscript technically sound, and do the data support the conclusions?

Reviewer #1: Yes

2. Has the statistical analysis been performed appropriately and rigorously? 

Reviewer #1: Yes

3. Have the authors made all data underlying the findings in their manuscript fully available?

Reviewer #1: Yes

4. Is the manuscript presented in an intelligible fashion and written in standard English?

Reviewer #1: Yes

5. Review Comments to the Author

Reviewer #1: Dear Authors,

Thank you very much for the opportunity to review this great study! I have a few suggestions for study improvement:

1) Overall:

- Please unify English spelling - sometimes fetus and other times foetus is used. Please double check for English grammar mistakes.

2) Abstract:

- Please provide 1-2 sentences regarding the rationale for choosing the superior gluteal nerve for your current study. Why is knowledge regarding this particular nerve important?

- First sentence of the conclusions - please rewrite adding information referring to the gestation. If you have partly divided the abstract into sections, please stick to it and present it as Introduction, materials and methods, results and conclusions.

- Please rewrite last two sentences of the conclusions - they are a repetition from the results. Instead provide information as to how to use your results in everyday medical practice.

3) Introduction:

- Paragraph two, second sentence - please specify what risk and add its estimated prevalence.

- Penultimate paragraph - the authors discuss the clinical implications associated with the superior gluteal nerve. However, information regarding the possible causes, mechanisms of injury or situations in which it can be damaged prenatally or in the neonatal period is scarce. Please add 1-2 sentences pertaining to this particular population, as those will be the patients that could potentially benefit from the study results. Please also reconsider the information regarding vaccinations into the gluteus muscles - the newborns are predominantly vaccinated (namely against hep. B and TB in some countries) by utilising the vastus lateralis muscle.

4) Material and methods:

- Please explain the "v-pl" abbreviation the first time it is used.

- The crown-rump length ought to be written down with a hyphen.

- Please provide the first and third quartiles instead of min-max (those can be described in the text), as the quartiles are more informative (the former can present outliers). The mean and median ought to be presented with one decimal place (SD with two) for mathematical accuracy.

- Please specify "visible developmental malformations" - did it involve only malformations of the abdomen, pelvis and lower limb or any malformation in general? How many foetuses were excluded?

- Please carefully review the paragraph describing the statistical methods used. Some parts have been submitted twice. Min and max values surely could not have been estimated but measured - please rewrite.

5) Results:

- The information regarding the measurement method from paragraph two ought to be presented in methods section, not results. Likewise, table 1 (characteristics of the analyzed material) with its calculated values ought to be presented in the results (calculations lead to results) with only a brief description in the materials section. Please move table 1 to the results section.

- Figure 14 and Figure 13 in paragraph 3 should not be referenced prior to figures 1-12. Good practice is to number the figures consecutively in the order they appear (as per journal guidelines).

- Sentence "it is divided into two types: tree form (...)" requires rewriting (especially its second part) - possibly split into two sentences.

- Please stick to the "tree type" - do not introduce another descriptor as "woody type" if it refers to the same entity. Same goes for Table 2.

- The 95%CI for the OR in Table 2 have to be corrected. Please rename the table (possibly into: "Table 2. Prevalence of superior gluteal nerve patterns in respect to sex and laterality.").

- Please unify the results as written with a dot (.) describing the decimal place of the estimates (not comas).

- Same as with table 1, please provide Q1 and Q3 values instead of min-max for table 3. Muscle abbreviations have to be explained in table footnotes. Please rename the table (possibly into: "Width of the superior gluteal nerve branches to the particular muscles in respect to sex and laterality.").

- Please rewrite "gluteal muscle medium" unerneath the table 4.

6) Discussion:

- The paragraph regarding the development of the lumbar spine is not relevant to the current study. Please delete.

- Clinical significance of the fetal superior gluteal nerve has to be expanded. Please relate it more to the fetal (possibly fetal lower limb surgery), neonatal and infant applications, as this is the population that could benefit from the current study. The embryological pattern of the nerve will be less relevant in an adult due to the growth of the axial skeleton, muscular variabilities and personal history (namely injuries) in the later life.

- There is no limitations section to the discussion. Please amend.

7) Conclusions:

- Please provide 1-2 sentences on how the results of the current study can specifically be used in everyday medical practice.

8) Please acknowledge the statisticians in the acknowledgement section, as well as the donors as per anatomical journals editors recommendations (https://onlinelibrary.wiley.com/doi/abs/10.1002/ca.23671).

9) Please change section headings into "Table legends" and "Figure legends." The tables were not submitted as supplementary files, but part of the main manuscript.

10) Figures:

- There is no number 3 (TFL) visible on the figure (though described in figure legends). It is also hard to see that the blue arrow points to the nerve to TFL. Please amend.

- Figure 2 - yellow arrow seems to aim more at gluteus medius than minimus.

- Please double check all figures against their footnotes - in many cases TFL is only listed, but not marked.

6. PLOS authors have the option to publish the peer review history of their article (what does this mean?). If published, this will include your full peer review and any attached files.

Reviewer #1: No

---

## [Author Response · Author response to Decision Letter 0]

2 May 2022

Reviewer #1: We have incorporated all of your suggestions into our revision, except part of suggestion 4 concerning decimal place for mathematical accuracy ("The mean and median ought to be presented with one decimal place (SD with two) for mathematical accuracy.").

After discussing with our statistician, we ask you to accept our version of the decimal places for the mean and median, because the number of decimal places depends on the accuracy of the measurement of the physical quantity. When measuring the length of biological structures (in millimeters), the actual average should be recorded to one decimal place. A measure of variation, i.e. standard deviation (SD), is always given in the same units with the same accuracy, so one decimal place is also enough.

Thank you for your help.

---

## [Decision Letter · Decision Letter 1]

30 May 2022

PONE-D-21-33849R1The morphometrical and topographical evaluation of the superior gluteal nerve in the prenatal period.PLOS ONE

Dear Dr. Derkowski,

Thank you for submitting your manuscript to PLOS ONE. After careful consideration, we feel that it has merit but does not fully meet PLOS ONE’s publication criteria as it currently stands. Therefore, we invite you to submit a revised version of the manuscript that addresses the points raised during the review process.

We look forward to receiving your revised manuscript.

Kind regards,

Mateusz Koziej, MD, PhD, DSc

Academic Editor

PLOS ONE

Journal Requirements:

Reviewers' comments:

Reviewer's Responses to Questions

**Comments to the Author**

1. If the authors have adequately addressed your comments raised in a previous round of review and you feel that this manuscript is now acceptable for publication, you may indicate that here to bypass the “Comments to the Author” section, enter your conflict of interest statement in the “Confidential to Editor” section, and submit your "Accept" recommendation.

Reviewer #1: (No Response)

2. Is the manuscript technically sound, and do the data support the conclusions?

Reviewer #1: Yes

3. Has the statistical analysis been performed appropriately and rigorously? 

Reviewer #1: Yes

4. Have the authors made all data underlying the findings in their manuscript fully available?

Reviewer #1: Yes

5. Is the manuscript presented in an intelligible fashion and written in standard English?

Reviewer #1: Yes

6. Review Comments to the Author

Reviewer #1: Dear Authors,

Many thanks for applying the suggested changes!

I have one final comment left - limitations of the study. Please add this section as the last paragraph of the discussion to show that you have fully endorsed the topic with any restrictions that could have presented on the way. Prominent medical databases like PubMed are very keen on articles having this section present.

7. PLOS authors have the option to publish the peer review history of their article (what does this mean?). If published, this will include your full peer review and any attached files.

Reviewer #1: No

---

## [Author Response · Author response to Decision Letter 1]

12 Jun 2022

Dear Reviewer,

 We have incorporated all of your suggestions into our revision. 

We have added section "limitations of the study" as the last paragraph of the discussion.

Thank you for your help.

 Kind regards

 Wojciech Derkowski, MD, PhD

---

## [Decision Letter · Decision Letter 2]

11 Jul 2022

PONE-D-21-33849R2The morphometrical and topographical evaluation of the superior gluteal nerve in the prenatal period.PLOS ONE

Dear Dr. Derkowski,

Thank you for submitting your manuscript to PLOS ONE. After careful consideration, we feel that it has merit but does not fully meet PLOS ONE’s publication criteria as it currently stands. Therefore, we invite you to submit a revised version of the manuscript that addresses the points raised during the review process.

Please make changes according to Reviewer 2

We look forward to receiving your revised manuscript.

Kind regards,

Mateusz Koziej, MD, PhD, DSc

Academic Editor

PLOS ONE

Journal Requirements:

Reviewers' comments:

Reviewer's Responses to Questions

**Comments to the Author**

1. If the authors have adequately addressed your comments raised in a previous round of review and you feel that this manuscript is now acceptable for publication, you may indicate that here to bypass the “Comments to the Author” section, enter your conflict of interest statement in the “Confidential to Editor” section, and submit your "Accept" recommendation.

Reviewer #1: All comments have been addressed

Reviewer #2: (No Response)

2. Is the manuscript technically sound, and do the data support the conclusions?

Reviewer #1: (No Response)

Reviewer #2: Yes

3. Has the statistical analysis been performed appropriately and rigorously? 

Reviewer #1: (No Response)

Reviewer #2: No

4. Have the authors made all data underlying the findings in their manuscript fully available?

Reviewer #1: (No Response)

Reviewer #2: Yes

5. Is the manuscript presented in an intelligible fashion and written in standard English?

Reviewer #1: (No Response)

Reviewer #2: Yes

6. Review Comments to the Author

Reviewer #1: (No Response)

Reviewer #2: Thank you for the opportunity of the reviewing this interesting article untitled: "The morphometrical and topographical evaluation of the superior gluteal nerve in the prenatal period." This article is well-written and the conclusion is clinically useful. I have one suggestion. In table 2 it is suggested to rename Fisher exact test for chi2 with Yates correction. This formula should be used when at least one cell of the table has an expected count smaller than 5. In authors analysis the outcome is similar to the Fisher test, however Yates cor. is more appropriate.

7. PLOS authors have the option to publish the peer review history of their article (what does this mean?). If published, this will include your full peer review and any attached files.

Reviewer #1: No

Reviewer #2: No

---

## [Author Response · Author response to Decision Letter 2]

21 Jul 2022

Dear Reviewer,

 We have incorporated all of your suggestions into our revision. 

We have changed in table 2 Fisher exact test for chi2 with Yates correction.

Thank you for your help.

 Kind regards

 Wojciech Derkowski, MD, PhD

---

## [Decision Letter · Decision Letter 3]

9 Aug 2022

The morphometrical and topographical evaluation of the superior gluteal nerve in the prenatal period.

PONE-D-21-33849R3

Dear Dr. Derkowski,

We’re pleased to inform you that your manuscript has been judged scientifically suitable for publication and will be formally accepted for publication once it meets all outstanding technical requirements.

Kind regards,

Mateusz Koziej, MD, PhD, DSc

Academic Editor

PLOS ONE

Additional Editor Comments (optional):

Reviewers' comments:

Reviewer's Responses to Questions

**Comments to the Author**

1. If the authors have adequately addressed your comments raised in a previous round of review and you feel that this manuscript is now acceptable for publication, you may indicate that here to bypass the “Comments to the Author” section, enter your conflict of interest statement in the “Confidential to Editor” section, and submit your "Accept" recommendation.

Reviewer #2: All comments have been addressed

2. Is the manuscript technically sound, and do the data support the conclusions?

Reviewer #2: (No Response)

3. Has the statistical analysis been performed appropriately and rigorously? 

Reviewer #2: (No Response)

4. Have the authors made all data underlying the findings in their manuscript fully available?

Reviewer #2: (No Response)

5. Is the manuscript presented in an intelligible fashion and written in standard English?

Reviewer #2: (No Response)

6. Review Comments to the Author

Reviewer #2: Authors have improved their manuscript. There is nothing more to improve. Congratulations. It was a pleasure to revise this manuscript.

7. PLOS authors have the option to publish the peer review history of their article (what does this mean?). If published, this will include your full peer review and any attached files.

Reviewer #2: No

---

## [Editor Report · Acceptance letter]

16 Aug 2022

PONE-D-21-33849R3 

The morphometrical and topographical evaluation of the superior gluteal nerve in the prenatal period.  

Dear Dr. Derkowski:

I'm pleased to inform you that your manuscript has been deemed suitable for publication in PLOS ONE. Congratulations! Your manuscript is now with our production department. 

Kind regards, 

on behalf of

Dr. Mateusz Koziej 

Academic Editor

PLOS ONE